# From Risk Assessment to Management: Cardiovascular Complications in Pre- and Post-Kidney Transplant Recipients: A Narrative Review

**DOI:** 10.3390/diagnostics15070802

**Published:** 2025-03-21

**Authors:** Thomas Beaudrey, Dimitri Bedo, Célia Weschler, Sophie Caillard, Nans Florens

**Affiliations:** 1Nephrology Department, Hôpitaux Universitaires de Strasbourg, 67000 Strasbourg, France; thomas.beaudrey@chru-strasbourg.fr (T.B.); dimitri.bedo@chru-strasbourg.fr (D.B.); celia.weschler@chru-strasbourg.fr (C.W.); sophie.ohlmann@chru-strasbourg.fr (S.C.); 2Inserm UMR_S 1109 Immuno-Rhumatology Laboratory, Translational Medicine Federation of Strasbourg (FMTS), FHU Target, Faculté de Médecine, Université de Strasbourg, 67000 Strasbourg, France; 3INI-CRCT (Cardiovascular and Renal Trialists), F-CRIN Network, 67000 Strasbourg, France

**Keywords:** cardiovascular disease, kidney transplant recipients, risk management, diagnostic strategies

## Abstract

Kidney transplantation remains the best treatment for chronic kidney failure, offering better outcomes and quality of life compared with dialysis. Cardiovascular disease (CVD) is a major cause of morbidity and mortality in kidney transplant recipients and is associated with decreased patient survival and worse graft outcomes. Post-transplant CVD results from a complex interaction between traditional cardiovascular risk factors, such as hypertension and diabetes, and risk factors specific to kidney transplant recipients including chronic kidney disease, immunosuppressive drugs, or vascular access. An accurate assessment of cardiovascular risk is now needed to optimize the management of cardiovascular comorbidities through the detection of risk factors and the screening of hidden pretransplant coronary artery disease. Promising new strategies are emerging, such as GLP-1 receptor agonists and SGLT2 inhibitors, with a high potential to mitigate cardiovascular complications, although further research is needed to determine their role in kidney transplant recipients. Despite this progress, a significant gap remains in understanding the optimal management of post-transplant CVD, especially coronary artery disease, stroke, and peripheral artery disease. Addressing these challenges is essential to improve the short- and long-term outcomes in kidney transplant recipients. This narrative review aims to provide a comprehensive overview of cardiovascular risk assessment and post-transplant CVD management.

## 1. Introduction

Kidney transplantation is associated with a long-term reduction in cardiovascular risk compared with wait-listed patients, but cardiovascular disease (CVD) remains a prominent cause of morbidity and mortality in kidney transplant recipients. Post-transplant CVD results from complex interactions between (1) traditional cardiovascular risk factors (i.e., diabetes, hypertension, dyslipidemia and tobacco), (2) specific factors linked with chronic kidney disease and transplantation such as mineral bone disease or graft dysfunction, and (3) previously known or hidden cardiovascular comorbidities. The spectrum of post-transplant CVD covers coronary artery disease, stroke, and peripheral artery disease, with some peculiar complications such as early post-transplant coronary artery disease, undetected despite pretransplant screening, and accelerated arteriosclerosis occurring in the transplant. We aimed to conduct a narrative review on the risk assessment and management of cardiovascular disease in the post-kidney transplantation period.

In this narrative review, we tried to deliver an overview of post-transplant cardiovascular risk and disease, starting with risk assessment. We first briefly describe the epidemiology of cardiovascular events before studying the field of traditional and specific cardiovascular risk factors. We also treat the specific and crucial screening of coronary artery disease before transplantation. In the second part, we were interested in the management of the previously described risk factors, with a comprehensive overview of cardiorenal medical therapies, immunosuppression, and vascular access management. Finally, in the third part, we focus on the management of the main cardiovascular diseases occurring after transplantation, especially coronary artery disease, cerebrovascular disease, atrial fibrillation, and peripheral artery disease.

The selection of the references did not follow a systematic review approach, but we previously defined a set of general criteria to identify studies relevant to the topic after a search in the PubMed database. We prioritized studies including observational data or clinical trials related to cardiovascular risk factors and diseases in the particular population of kidney transplant recipients. The authors selected them based on their general evaluation of relevance, methodology, and sample size as well as their contribution to understanding the field of post-transplant cardiovascular disease. We tried to capture a broad range of perspectives including both well-established studies and emerging research that provide new insights into the field.

## 2. Assessment of Cardiovascular Risk in Kidney Transplant Recipients

### 2.1. Epidemiology of Cardiovascular Events

#### 2.1.1. Cardiovascular Morbidity and Mortality

CVD is the leading cause of morbidity and mortality in chronic kidney disease [1,2], particularly among patients undergoing dialysis [3]. Kidney transplantation is the optimal treatment for kidney failure, as it is associated with improved survival compared with waitlisted patients [4]. Notably, much of this survival benefit is due to a reduction in cardiovascular mortality compared with dialysis [5,6]. Historically, CVD has been the primary cause of death among kidney transplant recipients, accounting for over 30% of deaths in the 1990s [7,8]. Today, CVD remains the leading cause of death in this population [9,10]; however, its incidence has significantly decreased [11,12], likely due to advancements in primary and secondary prevention strategies as well as improvements in treatment and despite the older median age at the time of waitlisting and transplantation. Notable disparities exist between continents, with CVD-related deaths accounting for approximately 20% in Europe compared with 25–30% in the United States [9] and in Australia and New Zealand [10]. These discrepancies are mainly attributable to inherent differential cardiovascular risk between high-income countries, with a CVD-related mortality consistently higher in the United States [13]. Furthermore, a high rate of sudden deaths, potentially attributable to cardiovascular causes, could lead to an underestimation of these figures [14].

CVD after kidney transplantation encompasses various conditions including coronary artery disease (CAD), heart failure, cardiac arrhythmias, and stroke. In the FAVORIT study, prevalent CVD was reported in 20% of patients five years post-transplantation [15]. According to a recent UK study, the cumulative incidence of CVD in naive recipients was 5.4% at five years and 14.3% at ten years [16], while in the U.S., it was reported as 8.3% at five years [17]. Compared with the general population, the relative risk of cardiovascular death in recipients aged 40 to 50 was 46 [18]. Additionally, CVD places a significant burden on national healthcare systems, with a growing number of cases and a median cost for post-transplant CVD-related hospitalizations [19].

The primary cause of CVD-related morbidity and mortality in kidney transplant recipients is CAD. Numerous studies have highlighted the incidence of myocardial infarction, ranging from 11.1% at three years post-transplant [20] to 14% after five years [15]. However, these data primarily pertain to U.S. recipients. A post hoc analysis of the multinational PORT study reported a cumulative incidence of CAD of 3.1% at one year and 7.6% at five years. These discrepancies are largely attributed to a higher incidence of CAD in North American patients compared with European recipients (approximately 6% at five years) [21]. Among waitlisted patients undergoing transplantation, the prevalence of CAD is reported to be as high as 20% [22], although the impact of CAD screening on post-transplant mortality remains unclear [23].

Heart failure with reduced ejection fraction (HFrEF) prevalence is not well-defined in waitlisted patients [24], but baseline systolic dysfunction appears to be associated with increased mortality [25]. While left ventricular ejection fraction and other sonographic parameters often improve post-transplantation [26], Lentine et al. reported an HFrEF incidence of 18.3% after transplantation, which was associated with reduced patient survival. Unfortunately, there are no data on the incidence of heart failure with the preserved ejection fraction (HFpEF) in kidney transplant recipients. Atrial fibrillation (AF), the most common cardiac rhythm disorder, is detected in 6% of kidney transplant recipients at the baseline and is associated with increased mortality [27]. The cumulative incidence of AF after transplantation is approximately 7.3% [28]. Pulmonary hypertension is another frequent cardiovascular complication in kidney transplant recipients [29], with a prevalence of around 20% according to small retrospective cohorts [30,31]. One prominent cause may be high cardiac output secondary to arteriovenous fistulas [32]. Valvular calcifications are observed in approximately 30% of recipients after one year of follow-up [33]. Although kidney transplantation reduces hospitalizations for valvular heart disease compared with waitlisted patients, its effect on disease progression remains unclear [34]. In a recent French cohort, recipients with prosthetic heart valves had decreased overall survival, with only mechanical valves being independently associated with mortality compared with bioprosthetic valves. Transcatheter aortic valve implantation appears to be a safe option for recipients with high-grade aortic stenosis [35]. Ventricular arrhythmias are detected in up to 30% of recipients [36] and may contribute to the high rate of sudden death observed in this population [14].

Accelerated arteriosclerosis should also be considered as a part of post-transplant cardiovascular disease. Arteriosclerosis in kidney transplants is defined as a vascular fibrous intimal thickening, according to the Banff classification [37], progressing after kidney transplantation and distinct from baseline donor lesions [38]. The pathophysiology of transplant arteriosclerosis remains unclear but probably results from two main mechanisms: hypertension and other cardiovascular risk factors can indeed induce these types of lesions [39], but the most recent hypothesis incriminates the alloimmune response related to donor-specific antibodies. In this study by Loupy et al. [40], risk factors for arteriosclerosis were donor parameters, inducing baseline lesions, cold ischemia time, recipient hypertension requiring medical therapy, and circulating donor-specific antibodies. In the same study, transplant arteriosclerosis was associated with graft failure and cardiovascular events, especially in the presence of donor-specific antibodies.

#### 2.1.2. Early Cardiovascular Events

The postoperative period carries a high risk of cardiovascular events, particularly CAD events and myocardial infarctions (MIs). Notably, Kasiske et al. demonstrated that the risk of MI was significantly higher in the postoperative period compared with waitlisted patients, with hazard ratios of 3.57 in deceased donor recipients and 2.81 in living donor recipients during the first three months post-transplant. Subsequently, the risk decreased, and transplantation became protective against MI [41]. In the study by Lentine et al. [20], the cumulative incidence of post-transplant MI was 4.3% at six months. CVD remains the leading cause of death in the initial months following kidney transplantation, with a mortality rate of 1.18 per 100 patient-years, as reported in the Australian study by Ying [10].

In an observational study by De Lima et al., 8.4% of recipients experienced a CAD-related event within the first year post-transplantation [42]. Similarly, Cheng et al., using Medicare data, reported incidences of 2.9% for MI and 2.6% for cardiac death [23]. An analysis of the U.S. Renal Data System found an MI incidence of 4.7% within the first month after transplantation [43]. Interestingly, Medicare data also showed a decline in the incidence of these outcomes over time. In Europe, the data are more limited, but the incidence of early coronary events appears to be lower. The PORT study reported a reduced incidence of such events [21], and a UK study using National Health Service data found the incidence of major adverse cardiovascular events (MACE) to be 0.9% within 90 days post-transplantation [44].

Despite these findings, each study indicated that pretransplant CAD screening, whether invasive or non-invasive, did not adequately predict or prevent early post-transplant coronary events. While risk factors for coronary events are well-defined [21], new strategies must be developed to prevent these early complications, which significantly contribute to the morbidity and mortality of newly transplanted recipients.

### 2.2. Cardiovascular Risk Factors

CVD in kidney transplant recipients is characterized by distinctive risk factors and atypical mechanisms. Both pre- and post-transplant factors contribute significantly to the development of heart failure, CAD, arrhythmias, and ultimately, cardiovascular mortality in this population. Traditional cardiovascular risk factors, such as diabetes, hypertension, dyslipidemia, and smoking, remain important but are exacerbated by transplant-specific factors. These include prolonged dialysis duration, impaired graft function, systemic inflammation from rejection episodes, post-transplant diabetes, and the effects of immunosuppressive therapies such as calcineurin inhibitors (e.g., tacrolimus-related diabetes risk and cyclosporine-induced hypertension) and corticosteroids (Figure 1).

#### 2.2.1. Traditional Risk Factors

Age remains a prominent risk factor, as in the general population [45]. Despite its high prevalence in transplant recipients, hypertension remains a significant contributor to CVD. A post hoc analysis of the FAVORIT trial by Carpenter et al. described a linear relationship between systolic blood pressure and the occurrence of CVD and death, with no association observed for diastolic blood pressure [46]. Several studies also demonstrated that high blood pressure increased the risk of graft loss [47,48].

Sex differences in cardiovascular risk are less well-studied. Liu et al. used a propensity score-matched cohort to show that women had a lower risk of MACE among kidney transplant recipients aged 50 or older with a waiting time of less than six years [49]. The PORT study added that men were at higher risk of CAD [21].

Regarding dyslipidemia, Roodnat et al. found that elevated serum cholesterol negatively impacted graft and patient survival [50]. The extension of the ALERT study formally demonstrated that reducing LDL cholesterol with fluvastatin reduced long-term cardiac death or nonfatal MI by 29% [51,52].

Diabetes mellitus is a major risk factor in kidney transplant recipients. Post-transplant diabetes mellitus (PTDM) was reported to occur in 9.1% of recipients in the cyclosporine era and was associated with poorer kidney outcomes [53]. Tacrolimus use significantly increased PTDM risk [54], which in turn mediated a higher risk of cardiovascular events [55]. However, pretransplant diabetes [56] and impaired glucose tolerance [57] also play a critical role. In a Canadian cohort of 5000 recipients, pretransplant diabetes was associated with a higher risk of MACE than both euglycemic and PTDM patients [58]. Additionally, diabetic kidney disease independently increased the risk of cardiovascular death compared with other histologic subtypes in the USRDS analysis.

Tobacco use is a significant risk factor for cardiovascular events, with a dose-dependent relationship between pack-years at transplantation and cardiovascular outcomes [59]. Incident smoking post-transplantation is also associated with poorer graft outcomes [60].

#### 2.2.2. Non-Classical Risk Factors

In addition to traditional risk factors, the PORT study identified several atypical risk factors specific to transplant recipients [21]. These include obesity, prolonged dialysis duration, and delayed graft function, which are associated with CAD events in the first year post-transplantation. Acute rejection, a panel reactive antibody > 10%, and graft dysfunction were linked to a higher risk of CAD within three years of follow-up.

Obesity exacerbates other risk factors such as diabetes and hypertension, with each five-unit increase in body mass index (BMI) raising the cardiovascular risk by 26% [61]. An increase in the waist-to-hip ratio post-transplantation is also associated with cardiovascular events [62], as is metabolic syndrome [63].

Dialysis duration is a critical factor, reflecting cumulative cardiovascular impairment, mineral bone disease, and calcium-phosphorus dysregulation. Each additional year of pretransplant dialysis increases the cardiovascular death risk by 14% [64]. A Swedish study of 889 recipients suggested that parathyroidectomy might reduce cardiovascular events incidence compared with medical therapy [65].

Delayed graft function (DGF) also impacts cardiovascular outcomes. Tapiawala et al. reported an increased rate of cardiovascular death among patients with delayed graft function [66].

High-flow arteriovenous fistula is associated with worse echocardiographic parameters [67], and fistula by itself is associated with heart failure after transplantation with a hazard ratio of 2.14 [68].

Acute rejection has been identified as an independent cardiovascular risk factor, even in the absence of graft dysfunction [21,69]. Similarly, proteinuria is associated with cardiovascular events [70], though its incidence remains unchanged despite the use of renin-angiotensin-aldosterone system (RAAS) blockers [71].

Kidney function directly influences cardiovascular outcomes. A post hoc analysis of the FAVORIT trial by Weiner et al. showed that each 5 mL/min/1.73 m^2^ decline in glomerular filtration rate (GFR) increased the MACE risk by 15%, with risks becoming more pronounced below a GFR of 45 mL/min/1.73 m^2^ [72].

Immunosuppressive therapy significantly contributes to cardiovascular risk. Calcineurin inhibitors are associated with dyslipidemia, hypertension, and PTDM [73], while corticosteroids contribute to bone mineral disease and metabolic syndrome. In contrast, belatacept was associated with lower PTDM rates [74], reduced proteinuria, and improved survival compared with calcineurin inhibitors [75]. mTOR inhibitors have not been proven to be effective in mitigating cardiovascular complications [73].

Other factors, including systemic inflammation [76] and left ventricular hypertrophy, also increase the cardiovascular risk. Left ventricular hypertrophy, often driven by anemia and hypertension, persists in some recipients despite improvement after transplantation [77,78].

### 2.3. Strategies for Screening for Coronary Artery Disease in Potential Kidney Transplant Recipient (PKTR)

#### 2.3.1. Pretransplant Coronary Artery Disease Screening: Why?

The goal of pretransplant cardiac evaluation is to identify high-risk patients who may benefit from specific management strategies prior to transplantation, thereby reducing the risk of MACE and early cardiovascular death post-transplant. This evaluation also helps to pinpoint patients for whom transplantation may be deemed too risky or contraindicated. Screening for undiagnosed CAD is particularly important due to its high prevalence in CKD patients [79,80] and its role as the leading cause of post-transplant mortality [12].

CKD patients often present with atypical or absent symptoms of CAD [81] such as less frequent chest pain compared with the general population [82,83]. This discrepancy is partially attributed to the high prevalence of diabetes and uremic neuropathy in this population. However, the benefits of screening must be weighed against the significant risks associated with invasive procedures like coronary angiography, which can lead to complications including hematoma at the puncture site, arterial injury, stroke, MI, and sudden cardiac arrest.

Recently, the European Society of Cardiology (ESC) 2024 guidelines for managing chronic coronary syndromes in the general population [84] introduced a stratification model categorizing patients into three risk groups, each benefiting from tailored screening approaches. However, CKD patients exhibit unique characteristics, such as a higher prevalence of CAD and elevated pre- and post-transplant cardiovascular risks, which diminish the applicability of these standard recommendations to the KTR population. Additionally, the performance of common screening tools is often suboptimal in CKD patients.

Unlike other surgical procedures, kidney transplantation carries the additional risk of graft dysfunction, particularly in the context of organ shortage. Consequently, the primary focus of pretransplant CAD screening should be on identifying patients who genuinely require optimization to reduce perioperative and long-term cardiovascular risks. At the same time, it is crucial to minimize unnecessary risks, delays, and costs for patients unlikely to benefit from intervention.

#### 2.3.2. Performance of the Main Available Tests in the CKD Population

Coronary CT angiography (CCTA) has proven diagnostic value in the general population but is less reliable in ESRD patients due to extensive coronary calcifications. A recent meta-analysis reported a sensitivity of 96% and specificity of 66% for detecting significant coronary stenosis [85], but also highlighted a high risk of false positives [86]. The development of computed tomographic fractional flow reserve (FFR-CT), a physiological simulation technique using routine CTA data to model coronary flow, pressure, and resistance, may improve diagnostic accuracy. This method helps assess the functional significance of coronary lesions, especially in cases where calcifications compromise CCTA interpretation [87]. FFR-CT has been significantly associated with post-transplant MACE. While a normal CCTA can reassure against macrovascular disease in patients with lower CAD likelihood, it does not address microvascular conditions. Conversely, a positive result may necessitate additional ischemia testing, often leading to delays and increased costs.

Coronary artery calcium scoring (CACS) uses adjusted thresholds for ESRD and dialysis-dependent ESRD populations. While helpful, its diagnostic performance is limited when used alone, with a sensitivity of 67% and specificity of 77% [86]. Combined with clinical data (e.g., fewer than three risk factors) and a CACS cutoff of <400, it offers a strong negative predictive value for CAD and correlates with the risk of MACE [88]. Dahl et al. [89] found that high-risk plaques and CACS were associated with MACE, but not with mortality, although calcified and low-attenuation plaque burdens were independently associated with both outcomes. Combining CACS with CCTA can improve the diagnostic accuracy. However, clinicians remain hesitant to use CCTA routinely, given concerns about radiation exposure and the potential toxicity of iodine contrast agents [90].

Stress cardiovascular magnetic resonance (CMR) is an effective tool for detecting myocardial ischemia but must be used cautiously in patients with ESRD due to potential risks associated with gadolinium-based contrast agents (GBCAs) [91]. While newer macrocyclic GBCAs may offer a safer alternative, their widespread use and validation in this population remain limited. Emerging techniques such as native T1 mapping and arterial spin labelling are currently under investigation and may provide contrast-free alternatives [92]. Dobutamine stress CMR, a non-contrast approach, has demonstrated high diagnostic accuracy for detecting CAD in potential kidney transplant recipients (PKTRs) [93,94]. However, large-scale studies are still needed to establish its role within broader screening strategies, particularly given the considerations around its cost and accessibility.

Single photon emission computed tomography (SPECT) provides functional assessment by evaluating myocardial perfusion and viability. However, diagnostic performance in CKD patients is limited due to factors such as triple-vessel disease, left ventricular hypertrophy (LVH), fibrosis, endothelial dysfunction, and volume overload [95]. Sensitivity and specificity for detecting stenosis >70% have been reported at 67% and 77%, respectively [96,97]. Despite these limitations, SPECT remains commonly used due to its accessibility, low risk, and strong predictive value for MACE [98,99].

Stress echocardiography (SE) offers advantages like accessibility, affordability, and the absence of radiation exposure. It provides a comprehensive evaluation of myocardial kinetics, blood flow, and valve function. However, diagnostic performance can be affected by concentric remodeling and eccentric hypertrophy, which may lead to false negatives [100]. SE shows a sensitivity of 76% and specificity of 88% for detecting CAD with stenosis >70% [96]. Compared with SPECT, SE demonstrates superior diagnostic and prognostic value and is recommended as a first-line test by the AHA 2022 statement. However, test selection should consider factors such as patient echogenicity, exercise capacity, and conditions like arrhythmia or hypo-/hypertension. Notably, SE and SPECT are comparable to coronary angiography in predicting all-cause mortality and MACE in kidney transplant recipients [101,102].

Positron emission tomography (PET-CT) is emerging as the most promising non-invasive diagnostic tool in the general population [103], offering lower radiation exposure and shorter test duration compared with SPECT. PET-CT not only evaluates CAD, but also assesses coronary flow reserve and myocardial blood flow (MBF), making it the gold standard for the non-invasive assessment of coronary microvascular dysfunction [104]. A recent single-center study compared PET-CT and SPECT in 393 pretransplant patients, finding PET-CT significantly associated with post-transplant MACE (hazard ratio [HR] 3.02 vs. HR 1.39 for SPECT) [105]. Although PET-CT’s diagnostic performance in the PKTR population has not been specifically studied, its promising results in the general population suggest similar utility. However, PET-CT’s high cost and limited availability warrant further research to define its role in pretransplant evaluation.

To enhance discrimination, imaging can be combined with biomarkers like high-sensitivity cardiac troponin T (hs-cTnT), which shows promise even in ESRD patients [106]. A recent study by Szramowska et al. evaluated 100 patients undergoing pre-kidney transplant screening with kinetic abnormalities, positive scintigraphy, or SE [107]. Testing for hs-cTnT and coronary angiography revealed significant stenosis in 50% of patients. An hs-cTnT cutoff value of 0.069 ng/mL demonstrated 61.4% sensitivity and 82.2% specificity for diagnosing obstructive CAD. Incorporating an age threshold of 52 years, the study proposed an algorithm where patients under 52 with hs-cTnT < 0.069 ng/mL could forgo further CAD evaluation, while those older or with hs-cTnT > 0.069 ng/mL would be referred for coronary angiography. Although this strategy may reduce unnecessary coronary angiographies, the study’s small, single-center design requires further validation.

Finally, commonly used risk scores, such as the Revised Cardiac Risk Index (RCRI) for 30-day perioperative cardiovascular risk [108] and the Framingham risk score for post-transplant MACE, show limited predictive value in CKD patients as they do not account for transplant-specific risk factors [109].

#### 2.3.3. Which Screening Strategy to Choose?

Screening strategies for asymptomatic patients on the kidney transplant waiting list have evolved significantly over the decades. According to the latest AHA 2022 statement [90], KTR presenting with symptoms or clinical signs of angina, heart failure, significant ischemia on ECG, left ventricular ejection fraction (LVEF) < 40%, kinetic abnormalities, arrhythmias, or valvular disease are classified as high risk. These patients should be referred directly for specialized cardiac evaluation, with coronary angiography considered as a first-line investigation.

For asymptomatic patients, new data now allow for the better identification of individuals who would benefit from revascularization. All candidates should undergo a minimum evaluation including a comprehensive risk factor assessment, symptom inquiry, ECG, and echocardiography. Notably, the KDIGO 2020 Guidelines [110] recommend echocardiography only for asymptomatic patients who have been on dialysis for more than two years and have risk factors for pulmonary hypertension. Despite these guidelines, echocardiography remains a widely used screening tool due to its affordability, accessibility, and absence of associated risks.

Following this initial evaluation, patients with no significant abnormalities on ECG or echocardiography—defined as no valvular disease, LVEF >40%, and no kinetic abnormalities—can be categorized as low risk. Low-risk criteria include being under 60 years of age, no diabetes, no history of cerebrovascular or peripheral artery disease, and less than five years on dialysis. These patients generally do not require further evaluation.

Patients who do not meet the low-risk criteria are classified as intermediate risk and should proceed with additional screening. Patients with a history of known CAD are treated as higher risk compared with those without CAD, and their management depends on prior coronary angiogram findings, LVEF, and clinical signs. Based on this assessment, a strategy involving either optimized medical treatment or non-invasive ischemia testing (+/− coronary angiography) can be indicated (Figure 2).

The AHA 2022 statement was evaluated in a retrospective observational study [111], which demonstrated that potential kidney transplant candidates (PKTCs) who were not recommended for further screening had a low revascularization rate and fewer adverse outcomes. However, the 2022 algorithm directed a higher percentage of patients for cardiac referral or screening compared with the AHA 2012 algorithm (73% vs. 53%).

Regarding ischemia screening, the AHA 2022 statement does not specify a gold standard test. However, they appear to favor stress echocardiography, likely due to its cost-effectiveness and potentially superior specificity compared with other imaging modalities.

## 3. Management of Post-Transplant Cardiovascular Risk

### 3.1. Treatment of Traditional Risk Factors in Kidney Transplant Recipients

Managing traditional risk factors in KTR is crucial and differs significantly from approaches used in chronic kidney disease (CKD) and dialysis patients (Table 1).

Hypertension: Calcium channel blockers are considered first-line treatment for hypertension in KTRs, as they have been shown to reduce the risk of graft loss (relative risk [RR] 0.75) compared with angiotensin-converting enzyme inhibitors (ACEis) [112]. However, ACEis should be prioritized in cases with significant proteinuria, given their renoprotective effects. Angiotensin receptor blockers (ARBs), beta-blockers, and diuretics are also valuable options depending on the patient’s clinical profile. The American Heart Association (AHA) recommends a target blood pressure of less than 130/80 mmHg for KTRs, as established in 2018 [113].

Dyslipidemia: Dyslipidemia should be addressed in all KTRs following the KDIGO 2013 Guidelines on lipid management [114]. In the ALERT study, fluvastatin therapy reduced the LDL cholesterol levels by 32% and achieved a significant 21% reduction in long-term MACE occurrence [51,52]. Ezetimibe can be used as a second-line therapy in combination with statins [115]. For hypertriglyceridemia, drug treatment is not currently recommended as per the KDIGO Guidelines. LDL cholesterol targets for KTRs are generally aligned with those for the general population and depend on cardiovascular risk. According to the ESC 2019 Guidelines, KTRs fall into high or very high-risk categories. Most recipients should aim for an LDL target of <0.7 g/L, while those at very high risk (e.g., GFR < 30 mL/min/1.73 m^2^, history of CVD, or diabetes with organ damage) should target <0.55 g/L [116]. Clinicians should note that cyclosporin is related to an increased systemic exposure to statins through interaction with cytochrome P450 3A4, p-glycoprotein, and organic anion transport polypeptide. However, the impact of tacrolimus on statin metabolism is less clear [117,118]. The American Heart Association still recommends reducing statins to the minimum available dosage in patients treated with calcineurin inhibitors, except for pravastatin and fluvastatin [119].

Diabetes: For pretransplant diabetes, insulin therapy is often necessary, particularly in the perioperative period [120]. Given their proven benefits in reducing cardiovascular risk [121] and slowing CKD progression [122], SGLT2 inhibitors should be considered first-line therapy for diabetes management in KTRs [123,124]. GLP-1 receptor agonists are another first-line option, supported by recent studies demonstrating their cardiovascular benefits [125,126] and tolerability in KTRs [127]. Additional options include metformin (with caution in patients at risk of lactic acidosis) and DPP-4 inhibitors. The KDIGO 2009 Guidelines recommend an HbA1c target of 7–7.5% in KTRs and advocate for aspirin use in secondary prevention or in primary prevention for recipients at high cardiovascular risk [128]. Furthermore, strategies such as early corticosteroid discontinuation [129] and switching to belatacept [74] may improve diabetes management and outcomes.

Smoking: Smoking cessation is a critical concern in KTRs. Smoking increases the risk of graft failure, cardiovascular events, and malignancies. Smoking cessation should be prioritized, and patients should be encouraged to quit during every clinical visit. First-line therapies include structured cessation programs and nicotine replacement therapies [130,131]. Second-line options include varenicline, which requires a dose adjustment based on GFR and bupropion, which should be used cautiously due to seizure risks.

**Table 1 diagnostics-15-00802-t001:** Management of traditional cardiovascular risk factors among kidney transplant recipients.

	Modality of Screening	Timing of Screening	Treatment	Target	Guidelines
Hypertension	Home blood pressure self-monitoring and measurement at office visit.	Blood pressure measurement at every visit	First-line: Calcium channel blockers, particularly after transplantation. Consider ARBs or ACE inhibitors if proteinuria is present. Alternative options: Thiazide diuretics, loop diuretics, beta-blockers.	BP < 130/80 mmHg For older recipients, consider < 140/90 mmHg.	AHA 2017 [113]
Dyslipidemia	Measurement of LDL-C, HDL-C, and triglycerides	Annually	All kidney transplant recipients should be treated with statins (per KDIGO 2013 Lipid Guidelines). Start with a low dose due to interactions with CNIs. Add ezetimibe to enhance LDL-C lowering if needed. For hypertriglyceridemia: Lifestyle changes recommended.	Most patients are at high or very high cardiovascular (CV) risk. LDL-C target < 0.7 g/L for patients with GFR < 30 mL/min/1.73 m^2^, history of CVD, or diabetes with organ damage. LDL-C target < 0.7 g/L in other transplant recipients.	KDIGO 2013 [114] 2019 ESC/EAS Guidelines [116]
Diabetes	Fasting plasma glucose, HbA1c, or oral glucose tolerance testing	Weekly during the first month, then at months 3, 6, 9, and 12, followed by annual assessments or when increasing the CNI dose.	Early initiation of insulin. Consider first-line therapies: SGLT2 inhibitors or GLP-1 receptor agonists. Other options: Metformin (if eGFR > 30 mL/min), DPP-4 inhibitors. Aspirin for secondary prevention or primary prevention in patients at high CV risk. Consider early corticosteroid discontinuation or a switch to belatacept to prevent post-transplant diabetes.	HbA1c 7–7.5%	KDIGO 2009 [128]
Smoking	Discuss tobacco use at each visit and assess whether the patient is willing to quit or needs assistance.	At every visit	Enroll patients in a smoking cessation program. Consider smoking cessation therapies: First-line: Nicotine replacement therapy. Second-line: Varenicline (dose-adjusted for GFR) or bupropion (note seizure risk when used with cyclosporine).	Smoking cessation	KDIGO 2009 [128]

### 3.2. Treatment of Non-Classical Risk Factors Management in Kidney Transplant Recipients

Several non-classical risk factors, including obesity, PTDM, proteinuria, and bone mineral disease, play a significant role in the cardiovascular outcomes of KTRs and are amenable to treatment.

Obesity: Although obesity is not a contraindication to kidney transplantation, it is associated with worse outcomes. Additionally, new transplant recipients are prone to post-transplant weight gain, primarily due to corticosteroid use, which further exacerbates cardiovascular risk [132]. Addressing obesity and weight gain involves lifestyle modifications, and bariatric surgery may be considered when necessary [133]. However, bariatric surgery has limitations, including the risk of enteric hyperoxaluria and nephrolithiasis, necessitating the development of alternative strategies to mitigate these risks. Emerging therapies, particularly GLP-1 receptor agonists like semaglutide, have shown promise in reducing cardiovascular risk and promoting weight loss in obese patients without diabetes [134], but evidence in KTRs remains limited.

Post-transplant diabetes mellitus (PTDM): Managing PTDM involves the early initiation of insulin therapy [120], standard diabetes treatments such as GLP-1 agonists and SGLT2 inhibitors, and optimizing immunosuppressive regimens. Switching to belatacept [74] or discontinuing corticosteroids [129] may also be beneficial in selected cases to improve glycemic control and reduce cardiovascular risk.

Proteinuria: Proteinuria in KTRs can be managed with RAAS blockers, although these drugs have not demonstrated significant benefits on graft or patient survival [71,135]. SGLT2 inhibitors offer potential benefits but lack robust data in the transplant population [122].

Bone mineral disease: Hyperparathyroidism is a leading contributor to bone mineral disease post-transplantation. Parathyroidectomy remains the gold standard treatment and has been shown to reduce cardiovascular events [65]. Medical treatments, such as cinacalcet, can lower the parathyroid hormone levels, but their effects on cardiovascular risk are unclear [74].

Graft dysfunction and acute rejection: Both graft dysfunction and acute rejection are significant risk factors for cardiovascular complications. Their management should adhere to standard practices including appropriate immunosuppressive regimens and vigilant monitoring [136,137,138].

### 3.3. The Role of Lifestyle Modifications

According to the KDIGO 2009 Guidelines, kidney transplant recipients should follow a “healthy lifestyle, with exercise, proper diet, and weight reduction as needed” [128]. These recommendations are obviously adapted from those concerning the general population, according to the 2019 ACC/AHA Guidelines, lifestyle factors can indeed influence and reduce cardiovascular risk in the general population [139]. Concerning nutrition and diet, patients should prioritize vegetables such as fruits, legumes, nuts, and whole grains as well as fish intake. Saturated fat should be replaced by mono- or polyunsaturated fats. The amount of cholesterol and salt consumption should be reduced. A high potassium intake could also be encouraged in the general population but should be carefully evaluated in kidney transplant recipients. Finally, as part of a healthy diet, the intake of trans fats, processed meats, refined carbohydrates, and sweetened beverages should be minimized. Patients should also be encouraged, if able, to routinely perform at least 150 min of moderate-intensity or 75 min of high-intensity physical activity per week. Lifestyle changes also imply weight loss, tobacco weaning, and alcohol avoidance. They participate in a global reduction in cardiovascular disease in the general population, mainly through the reduction in blood pressure and LDL-cholesterol levels, and the improvement in glycemic control. They also participate in reducing the BMI and are the baseline treatment for obesity [139]. The 2019 KDIGO Guidelines also state that there is no reason to believe that a healthy lifestyle and a proper diet do not prevent cardiovascular complications as in the general population [128].

Several randomized control trials have analyzed the impact of physical activity in the field of kidney transplantation, but are globally underpowered to show a real difference in the improvement in cardiovascular risk factors. Most of these are centered on the improvement in physical capacity and kidney function. A meta-analysis by Zhang et al. concerning exercise intervention in kidney transplant recipients and including 16 randomized control trials revealed a positive impact of exercise on renal function and physical performance, with a lower creatinine and improved VO2 peak and 6-min walk test [140]. However, a randomized control trial of 96 patients did not show significant improvement in cardiovascular risk, evaluated according to the Framingham equations, in the exercise arm in the first year after transplantation [141].

Concerning the impact of dietary changes, they were associated in several clinical trials to better kidney function and overall survival. In a randomized clinical trial including 632 kidney transplant recipients, the Mediterranean diet was associated with better graft function [142]. Another trial including 632 recipients determined that the DASH diet was associated with better graft function and lower all-cause mortality [143]. However, a recent systematic review regarding the dietary interventions for body weight management did not demonstrate any effect due to a lack of significant studies in chronic kidney disease [144] and kidney transplant recipients [145]. Unfortunately, the impact of dietary changes on specific cardiovascular outcomes are rarely depicted in the literature. In summary, if some specific diet styles are associated with better outcomes, the impact of dietary interventions remains to be proven.

The proper impact of physical activity and dietary changes remains unclear on specific cardiovascular risk factors and outcomes in kidney transplant recipients, mostly due to unadapted designs in clinical studies. However, due to their large beneficial impact in the general population, and based on the available literature, we can conclude that lifestyle changes are formally indicated in all kidney transplant recipients, regardless of their cardiovascular risk.

### 3.4. RAAS Inhibitors in Kidney Transplant Recipients

RAAS inhibitors, including ACEis and ARBs, are widely recognized for their role in reducing proteinuria, controlling hypertension, and slowing the progression of CKD. In KTRs, the use of RAAS inhibitors is particularly compelling due to the high prevalence of hypertension and proteinuria, both of which are associated with poor graft outcomes and increased cardiovascular morbidity. However, the utility of RAAS inhibitors in improving long-term graft survival and overall patient outcomes remains debated due to conflicting evidence from clinical trials and observational studies.

Proteinuria is a strong predictor of graft loss in KTRs. RAAS inhibitors effectively reduce proteinuria by decreasing intraglomerular pressure through efferent arteriolar vasodilation. This mechanism not only reduces protein excretion, but also mitigates glomerular damage. For example, Brenner et al. demonstrated in an RCT involving 1513 patients with diabetic nephropathy that losartan reduced proteinuria by 35% and cardiovascular events by 16%, highlighting its renoprotective and cardioprotective effect [146]. Similarly, in an RCT by Midtvedt et al., lisinopril effectively reduced proteinuria in hypertensive KTRs, though nifedipine demonstrated a more pronounced improvement in GFR, emphasizing the nuanced effects of different antihypertensive strategies [147].

Observational studies have also explored the impact of RAAS inhibitors on the long-term outcomes of KTRs. Heinze et al. reported a significant 35% reduction in graft loss risk (HR 0.65) and a 39% reduction in mortality (HR 0.61) in KTRs treated with RAAS inhibitors [148]. These findings suggest a beneficial effect on both graft longevity and patient survival, likely mediated by antiproteinuric and antihypertensive mechanisms. However, contrasting results were presented by Opelz et al. in the Collaborative Transplant Study (CTS), which analyzed 17,209 KTRs and found no significant improvements in graft survival (HR 1.05) or patient survival (HR 1.01) among those receiving ACEis or ARBs. Subgroup analyses of high-risk patients, such as those with diabetic nephropathy or hypertension, also failed to demonstrate meaningful benefits, raising questions about the generalizability of positive findings [149].

Further complicating the narrative are the results from more recent investigations, such as Knoll et al., which evaluated ramipril in a multicenter RCT of 213 KTRs with proteinuria. The study found no significant differences in key outcomes, including doubling of serum creatinine, ESRD, or mortality, compared with the placebo. Additionally, adverse effects such as hyperkalemia were more common in the ramipril group, underscoring the potential risks of RAAS blockade [150]. Similarly, Cheungpasitporn et al., in a meta-analysis of 20,024 KTRs, concluded that RAAS inhibitors offered no significant reduction in allograft loss or mortality, emphasizing the heterogeneity of the study designs and patient populations [151].

Despite these challenges, the potential cardioprotective and anti-inflammatory effects of RAS inhibitors remain of interest. Kovarik et al. demonstrated that ACEis suppressed systemic angiotensin II levels and promoted the alternative RAAS pathway, generating anti-inflammatory peptides like angiotensin-(1–7). These effects could theoretically mitigate chronic allograft nephropathy and cardiovascular complications. However, the phenomenon of “ACE escape”, where intrarenal angiotensin II synthesis persists despite systemic RAAS blockade, raises questions about the long-term efficacy of these agents [152].

In conclusion, while RAAS inhibitors effectively reduce proteinuria and may offer cardiovascular protection (Table 2), their role in improving graft and patient survival in kidney transplant recipients remains inconclusive. The conflicting evidence underscores the need for individualized treatment approaches, the careful monitoring of potential side effects, and the identification of patient subgroups most likely to benefit. Future research, particularly well-designed, large-scale RCTs, is essential to clarify the long-term benefits and risks of RAAS inhibitors in this unique population. For now, their use should be guided by patient-specific factors, with an emphasis on balancing the potential benefits with the known risks.

**Table 2 diagnostics-15-00802-t002:** Summary of key studies on RAS inhibitors in kidney transplant recipients.

Study (Year)	Population (N)	Design	Key Findings	Reference
Midtvedt et al. (2001)	72	RCT	Nifedipine improved GFR by 20%; lisinopril reduced proteinuria.	[147]
Heinze et al. (2006)	1513	Observational cohort	RAS inhibitors reduced graft loss risk (HR 0.65) and mortality (HR 0.61).	[148]
Opelz et al. (2006)	17,209	Observational cohort	No significant improvement in graft (HR 1.05) or patient survival (HR 1.01).	[149]
Knoll et al. (2016)	213	RCT	No significant effect on ESRD, mortality, or creatinine doubling in KTRs.	[150]
Cheungpasitporn et al. (2016)	20,024	Meta-analysis	No reduction in allograft loss or mortality; heterogeneous study designs.	[151]
Kovarik et al. (2019)	48	Cross-sectional	ACEis promoted anti-inflammatory RAS pathways but showed intrarenal escape.	[152]

RAS: renin-angiotensin system, ACEis: angiotensin-converting enzyme inhibitors, ARBs: angiotensin receptor blockers, RCT: randomized controlled trial, KTRs: kidney transplant recipients, GFR: glomerular filtration rate, ESRD: end-stage renal disease.

### 3.5. Emerging Therapies to Manage Cardiovascular Risk

Emerging therapies for cardiorenal protection in kidney transplant recipients, including SGLT2 inhibitors, finerenone, and semaglutide, represent promising strategies to mitigate the high cardiovascular and renal risks in this population. SGLT2 inhibitors, such as empagliflozin and dapagliflozin, have shown significant benefits in improving glycemic control, reducing albuminuria, and preserving allograft function. They achieve this by reducing glomerular hyperfiltration and intraglomerular pressure while exerting anti-inflammatory and antifibrotic effects [153,154].

A pivotal study leveraging the TriNetX database has provided compelling real-world evidence regarding the efficacy of SGLT2 inhibitors in diabetic kidney transplant recipients (KTRs). This study analyzed a cohort of 50,120 diabetic KTRs, and after propensity score matching, compared the outcomes between 1970 SGLT2 inhibitor users and an equal number of matched non-users with a median follow-up of 3.4 years. SGLT2 inhibitor users exhibited a remarkable reduction in all-cause mortality (adjusted hazard ratio [aHR]: 0.32, *p* < 0.001), MACE (aHR: 0.48, *p* < 0.001), and major adverse kidney events (MAKE; aHR: 0.52, *p* < 0.001) [123]. Further subgroup analyses revealed that these benefits extended across various demographic and clinical profiles including age, sex, baseline kidney function, and comorbidity burden. Importantly, the study also confirmed the safety profile of SGLT2 inhibitors in this population, with no significant increase in genitourinary infections or acute kidney injury. The observed reduction in albuminuria and attenuation of diabetic nephropathy progression underscores the nephroprotective potential of these agents [123]. However, these results should be interpreted with caution, since only 44% of the participants in this study were treated using RAAS inhibitors. All randomized controlled trials performed in non-transplanted CKD patients [121,122] compared the RAAS blockers-SGLT2 inhibitors’ association with RAAS blockers alone, and a similar design should be applied to the transplant population. It is rather reassuring that no significant difference in the effect size of the SGLT2 inhibitors was described in this study between the subgroups with or without RAAS inhibitor treatment.

These findings strongly advocate for the integration of SGLT2 inhibitors into the therapeutic regimen for diabetic KTRs. With their demonstrated ability to significantly reduce mortality and adverse cardiac and kidney events, these agents hold transformative potential for improving the post-transplant outcomes. Future studies, including randomized controlled trials, are essential to further elucidate the long-term safety and efficacy of these agents in diverse KTR populations.

Finerenone, a nonsteroidal mineralocorticoid receptor antagonist, offers additional nephroprotection by mitigating inflammation and fibrosis. Preclinical and early clinical data suggest that it may prevent calcineurin inhibitor nephrotoxicity and improve cardiovascular outcomes in KTRs [155]. Meanwhile, semaglutide, a GLP-1 receptor agonist, has demonstrated weight reduction and glycemic control benefits in KTRs with diabetes or metabolic syndrome, with no significant adverse impact on graft function in small studies. Its cardioprotective and nephroprotective effects make it a valuable addition to the therapeutic arsenal [156].

Together, these therapies hold the potential to reshape the management of KTRs, improving graft longevity and patient survival while addressing the dual challenges of cardiovascular and renal disease. However, further randomized controlled trials are needed to validate their long-term safety and efficacy in this unique population [157].

### 3.6. Challenges of Immunosuppression from a Cardiovascular Perspective

#### 3.6.1. T-Cells and Their Cardiovascular Impact

While rejection is clearly associated with an increased cardiovascular risk [158], the role of T-cell immunosuppression in this context remains a matter of debate in the literature. T lymphocytes are central to the pathogenesis of hypertension, vascular remodeling, and atherosclerosis [159,160]. The impact of specific T-cell subtypes is an area warranting further investigation. For instance, low CD4 counts, as observed in HIV patients, have been linked to elevated cardiovascular risk [161]. Among the T-cell subtypes, CD4 Th1 T-cells and CD8 T-cells appear to promote atherosclerosis, whereas Th2 T-cells exert protective effects [162]. Similarly, regulatory T-cells (T-Reg) not only provide protection against atherosclerosis, but may also mitigate vascular lesions following cardiac or kidney injury [159,160,162,163,164].

The potential cardiovascular impact of T-cell-depleting induction therapies in transplant recipients has been highlighted, particularly in a small retrospective study [165]. These therapies are thought to shift T-cell subsets, reducing the CD4 T-cell counts while promoting the expansion of late-stage differentiated CD8 T-cells, which may contribute to cardiovascular risk [166]. However, larger-scale observational data, such as a cohort of 162,998 KTRs from the United States Renal Data System (USRDS), provide a more reassuring perspective, suggesting no significant cardiovascular harm associated with T-cell depletion [167].

Despite these findings, the cardiovascular effects of T-cell modulation remain an unresolved issue. Further research is essential to clarify the mechanisms through which T-cell subsets influence cardiovascular outcomes in transplant recipients and guide clinical strategies for balancing immunosuppression with cardiovascular risk management.

#### 3.6.2. Off-Targets Adverse Effects of Maintenance Therapy: A Personalized Strategy

Maintenance therapy in solid organ transplantation typically relies on a regimen centered on calcineurin inhibitors (CNIs) combined with mycophenolate mofetil (MMF) and corticosteroids [128]. The cardio-renal off-target effects of these immunosuppressive therapies are summarized in Table 3.

#### 3.6.3. Addressing CNI Toxicity

While CNIs are highly effective in preventing rejection, they are associated with significant off-target effects, particularly metabolic complications such as hypertension, diabetes, and dyslipidemia [168]. Despite their experimental mitochondrial protective effects on ischemia-reperfusion injury [169], their cardiotoxicity and endothelial toxicity are increasingly recognized. These toxicities contribute to accelerated atherosclerosis, cardiac remodeling, and chronic cardiovascular-kidney disorder (CCKD), especially in kidney transplant recipients (KTRs) [168,170,171].

Therapeutic drug monitoring (TDM) remains essential for personalizing tacrolimus dosages to optimize immunosuppressive efficacy while minimizing adverse effects [6]. Emerging biomarkers, such as the torque teno virus levels, show promise for guiding immunosuppression and enhancing therapeutic precision [172].

#### 3.6.4. mTOR Inhibitors: A Perilous Alternative

Experimental studies suggest that mTOR inhibitors may slow the progression of atherosclerosis, although they frequently induce dyslipidemia [73]. While mTOR inhibitors may address allograft vasculopathy in heart transplant recipients [173], their safety and efficacy as CNI replacements in kidney transplantation remain questionable. A meta-analysis of 2323 patients showed a nonsignificant trend toward increased new-onset diabetes after transplantation (NODAT) and significant increases in hypercholesterolemia, acute rejection, proteinuria, and anemia [73].

#### 3.6.5. Belatacept: A Promising Alternative

Belatacept, a costimulation blocker, has emerged as a promising alternative to CNIs in kidney transplantation. The BENEFIT study demonstrated belatacept’s advantages in improving patient and graft survival while preserving renal function and slowing the decline in GFR [174]. Experimental studies also suggest that belatacept attenuates progressive renal injury, pressure overload-induced heart failure, and accelerated atherosclerosis [175,176,177].

In clinical trials, belatacept was associated with fewer cardiac disorders than cyclosporine (2% vs. 12%) in a five-year extension study [178]. In the seven-year extension, serious vascular disorders were less frequent with belatacept (3.5%) compared with cyclosporine (5.4%) [179]. Additionally, belatacept, alone or combined with minimized-dose tacrolimus, reduced the incidence of PTDM compared with tacrolimus alone (1.7%, 2.2%, and 3.8%, respectively) [74]. These findings suggest belatacept as a valuable strategy, particularly for high-risk cardiovascular patients, warranting further clinical trials to refine its use, but weighed against the risk of infection in some patients.

#### 3.6.6. Early Steroid Withdrawal: A Beneficial Strategy for Select Patients

Corticosteroid withdrawal after kidney transplantation is a common strategy in the United States, with approximately 30% of patients following a steroid-free regimen one year post-transplant [180]. A five-year randomized controlled trial showed reduced PTDM incidence and improved lipid profiles with early steroid withdrawal (ESW), though there was no significant impact on cardiovascular mortality. However, ESW increased biopsy-proven cellular rejection, albeit without affecting graft survival [181].

A retrospective study in deceased-donor KTRs reported better overall survival, graft survival, and fewer cardiac, metabolic, and bone complications in patients receiving reduced corticosteroid doses [182]. Conversely, a large U.S. registry found that in patients with DGF, corticosteroid discontinuation was associated with higher rejection rates and worse graft survival [183].

Bae et al. analyzed 210,086 adult deceased-donor KTRs, including 26,248 retransplant recipients, and reported mixed results. Among all recipients, ESW had no significant difference in acute rejection, a slightly higher hazard of graft failure, and slightly lower mortality compared with continuous steroid maintenance. However, retransplant recipients experienced higher acute rejection and graft failure risks with ESW [184].

A propensity score-matched study of 304 older KTRs (≥65 years) revealed that ESW significantly reduced the PTDM rates at one and three years compared with continuous steroid immunosuppression (22.36% vs. 30.37% and 34.89% vs. 44.29%, respectively). ESW also correlated with lower myocardial infarction rates at five years (6.75% vs. 14.39%) and fewer infection-related complications, suggesting improved patient and graft survival in this population [185].

While ESW shows promise for carefully selected patients, identifying reliable biomarkers and conducting targeted studies are urgently needed to refine corticosteroid minimization strategies in kidney transplantation.

### 3.7. Vascular Access Management After Transplantation: What Is the Best Option?

The creation of an arteriovenous fistula (AVF) for hemodialysis is a cornerstone in the management of ESRD [186]. While it is preferred for its long-term durability and lower risk of infection compared with central venous catheters or grafts, AVF creation significantly impacts the cardiovascular dynamics and poses unique challenges for KTRs [187].

#### 3.7.1. Cardiovascular Implications of AVF

AVFs, due to their high-flow, low-resistance nature, increase the venous return and cardiac output [32]. This additional strain on the heart can exacerbate pre-existing cardiovascular conditions. Increased cardiac output and pressure changes can contribute to pulmonary hypertension and left ventricular hypertrophy, while heightened preload and afterload may worsen heart failure, particularly in patients with reduced left ventricular function [188,189]. Additionally, elevated myocardial oxygen demand may exacerbate ischemia in individuals with coronary artery disease, further amplifying cardiovascular risk [190].

Notably, some of these cardiovascular effects may be reversible with AVF flow reduction in carefully selected patients. Studies have shown that reducing AVF flow significantly decreases the left ventricular mass, end-diastolic volume, end-diastolic diameter, and left atrial volume, with corresponding improvements in diastolic function. Reductions have also been observed in right ventricular diameter and right atrial volume, alongside a notable decrease in the estimated pulmonary artery systolic pressure [191,192].

#### 3.7.2. AVF Management Post-Transplantation

The management of AVFs after kidney transplantation is a topic of ongoing debate (Table 4). Approximately 50% of KTRs return to dialysis within a decade, highlighting the importance of maintaining AVF patency [192]. A retrospective study in Slovenia involving 626 KTRs found that 23.9% used AVF for delayed graft function, 8.4% for graft failure, and 5.4% for plasma exchanges. However, 29% experienced AVF-related complications including aneurysms, thrombosis, high-flow AVFs (HF-AVFs), distal hypoperfusion, venous hypertension, arm trauma, or pain [193]. Another study by Manca et al. reported spontaneous AVF closure in 167 out of 365 KTRs, while 42 patients required surgical intervention due to complications, emphasizing the risk of spontaneous thrombosis post-transplantation [194].

**Table 4 diagnostics-15-00802-t004:** HF-AVF management and cardiovascular and kidney outcomes.

Study	Design	Patients (N) Period	Intervention	Echographic Findings	Clinical Outcomes	Ref.
Stoumpos et al.	Observational study	1330 (2010–2020)	None, analysis of existing AVFs	Increased left ventricular mass, elevated cardiac output	Higher risk of new-onset heart failure post-transplant (adjusted Hazard Ratio 2.14)	[68]
Rao et al.	Randomized controlled trial	64 (2013–2017)	AVF ligation	Decrease in left ventricular mass, atrial volumes, and NT-proBNP levels	Improved cardiac remodeling, no impact on eGFR	[195]
Hetz et al.	Randomized controlled trial	28 (2013–2018)	Prophylactic ligation	Reduction in cardiac volumes and pulmonary systolic pressures	Prevention of high-output heart failure, decrease of NT pro BNP, no serum creatinine differences.	[196]
Keller et al.	Observational Prospective study	49 (2013–2015)	None, comparative analysis	Higher LVEDD and LVESD in high-flow AVF compared with normal-flow AVF	Associated with hemodynamic abnormalities	[67]
Janeckova et al.	Observational study	40 (2018–2023)	Flow reduction/ligation	Significant reduction in arterial flow and cardiac output post-ligation	88.3% primary patency after flow reduction, lower dyspnea according NYHA grade III, lower creatininemia and improve GFR	[197]

HF-AVFs—defined as a flow of 2 L/min or exceeding 20% of cardiac output—are relatively common post-transplantation, with a prevalence of 43–69% among functional AVFs in KTRs [67,197]. These HF-AVFs are associated with pathological cardiac remodeling and complications, as demonstrated by observational studies from Stoumpos et al. and Keller et al. [67,68,197].

#### 3.7.3. Evidence for AVF Ligation

RCTs by Rao et al. and Hetz et al. showed that AVF ligation can significantly reduce the left ventricular mass, improve cardiac remodeling, and prevent high-output heart failure without adversely affecting renal function [195,196]. However, these trials lack data on mortality and MACE. A study by Janeckova et al. uniquely highlighted the impact of AVF on both kidney and graft function, offering valuable insights into the cardiorenal interplay, though these findings require confirmation through larger RCTs [197].

A systematic review by Yasir et al. confirmed that ligating symptomatic AVFs in high-output heart failure is both safe and effective. The review also identified an AVF flow-to-cardiac output ratio >0.3 as a predictive marker for acute heart failure risk [198].

#### 3.7.4. Individualized AVF Management

These findings underscore the importance of individualized AVF management for KTRs. While preserving AVF patency is crucial for patients with unstable graft function, ligation should be considered in cases of HF-AVFs to improve cardiovascular outcomes. Clinicians must carefully balance the need to maintain vascular access for future dialysis with the potential cardiovascular benefits of AVF closure, tailoring decisions to the specific needs of each patient.

## 4. Management of Post-Transplant Cardiovascular Disease

### 4.1. Management of Coronary Artery Diseases in Kidney Transplant Recipients

The benefit of early revascularization in asymptomatic KTRs remains a debated issue within the scientific community, particularly given the inconsistent results observed over recent decades. Current evidence suggests no significant reduction in the risk of MACE or post-transplant mortality. This debate is increasingly relevant considering the advancements in therapies, diagnostic techniques, monitoring, and a deeper understanding of the pathophysiology of CAD. Studies advocating widespread revascularization for asymptomatic CAD in CKD patients are often small, retrospective, or outdated, making them incompatible with modern treatment strategies and tools [199]. No recent randomized prospective studies have conclusively demonstrated a benefit to this approach.

The ISCHEMIA-CKD trial [200] provided pivotal insights. It included patients with an estimated glomerular filtration rate (eGFR) < 30 mL/min/1.73 m^2^ who tested positive for ischemia (via SPECT, stress echocardiography, or exercise testing with moderate to severe results). Participants were randomized to either an invasive strategy (angiography ± revascularization) or conservative management (medical therapy and monitoring) and followed for a median of 2.2 years. Among those undergoing angiography, 57% had multivessel disease, 26% had no significant lesions, and 50% were revascularized. However, the primary outcome (cumulative incidence of nonfatal myocardial infarction and death) was 36% in both groups, demonstrating that invasive strategies do not improve outcomes compared with optimized medical therapy. This finding was supported by a post hoc analysis in KTRs [201].

When revascularization is indicated, the 2024 ESC Guidelines [84] recommend coronary artery bypass grafting (CABG) over percutaneous coronary intervention (PCI) for multivessel disease, particularly in CKD patients. Although PCI carries a lower short-term risk of vascular complications, death, and stroke, CABG offers superior long-term outcomes, with lower rates of myocardial infarction and repeat revascularizations [199,202,203].

An optimized medical strategy remains the cornerstone of CAD management. This includes antiplatelet agents, RAAS inhibitors, and beta-blockers where applicable. The use of statins in this population has been debated. Large randomized controlled trials in hemodialysis patients showed no significant reduction in cardiovascular risk with statin therapy, and a subgroup analysis from the SHARP study found no benefit of lowering LDL cholesterol in patients with a GFR < 30 mL/min [204]. Nevertheless, combining medical therapy with lifestyle modifications remains at least as effective in reducing MACE and improving survival, emphasizing the role of microvascular disease, which cannot be directly assessed or treated via coronary angiography.

The necessity of CAD screening itself has come under scrutiny. A retrospective study by Lee et al. [205] of 22,687 CKD patients found that transplant candidates underwent 5–6 times more non-invasive ischemia tests and 16% more coronary angiograms than dialysis patients, but had lower revascularization rates. No difference in one-year post-transplant mortality was observed between the screened and unscreened patients. However, screened patients experienced longer waitlist times (median delay of 599 days), attributed to comorbidities, the need for additional tests, delays from dual antiplatelet therapy, and logistical or financial burdens [206]. Nonetheless, these screenings allowed for better MACE risk stratification, aiding clinicians in tailoring management strategies.

No study to date has demonstrated a reduction in cardiovascular risk through screening. The ongoing CARSK study [207], a randomized trial, aims to test the hypothesis that discontinuing CAD screening after waitlisting is not inferior to standard MACE screening. It will also evaluate the transplant rates, safety, and cost-effectiveness.

The early post-transplant period poses the highest cardiovascular risk, with a 4.3% incidence of MI within the first six months [20]. Cheng et al. [23], in a quasi-experimental study, found no correlation between CAD screening and reduced MACE or early mortality, especially within the critical 30-day post-transplant period. Interestingly, some data even suggest that screening may increase event risks during this window, potentially due to confounding factors like inherent high risk among the screened patients. Randomized trials comparing screening and non-screening groups are needed to address these biases, including those identified by Kopparam et al. [208], which highlighted the lack of outcomes data for screened patients who were not selected for transplantation.

The inability to demonstrate a benefit of screening for early perioperative risks may also reflect the influence of immediate post-transplant factors. Inflammatory markers released during graft reperfusion, such as galectin-3 [209] and IL-33 [210], have been implicated in cardiac remodeling following acute kidney injury (AKI). Evidence from various populations (e.g., post-cardiac surgery [211], hospitalized patients [212], CKD patients [213]) shows that AKI increases cardiovascular risk. This “butterfly effect” describes how acute kidney insults can lead to long-term cardiovascular remodeling [170], partially explaining the increased post-transplant risk and the limited utility of screening. Other contributors include high-dose immunosuppressive therapy and de novo diabetes, which may play as significant a role as pretransplant cardiovascular risk factors in explaining post-transplant outcomes.

Cardiac screening for asymptomatic CAD has not demonstrated clear benefits in improving survival or reducing the incidence of MACE after transplantation. Similarly, many coronary angiograms performed in this context could likely be avoided, as the benefits of revascularization in these patients remain uncertain. Therefore, it is imperative to refine the current screening algorithm by integrating novel biomarkers and conducting robust, well-designed studies to provide stronger evidence and potentially reshape large-scale management strategies. The current stagnation in this area is largely due to insufficient data and the absence of universally accepted recommendations. Additionally, the contribution of cardiovascular risk carried by the graft itself remains undefined in this population and warrants further investigation.

### 4.2. Management of Cerebrovascular Disease in Kidney Transplant Recipients

The risk of stroke progressively increases in individuals with advanced CKD and those undergoing dialysis compared with the general population. Additionally, albuminuria independently contributes to this heightened risk, regardless of blood pressure and diabetes status [214]. The pathophysiology is complex, involving traditional cardiovascular risk factors, dialysis-related factors, and non-traditional risk factors [215] such as increased blood–brain barrier permeability [216]. Emerging research suggests that uremic toxins may play a critical role in stroke pathogenesis, presenting a promising avenue for further investigation [217].

The reported prevalence of stroke ranges from 1.3% to 8%, with ischemic strokes being more common (63–91.4%) than hemorrhagic strokes. Incidence rates vary between 5.26 per 1000 person-years to 5.34 per 1000 person-years, depending on the study design and follow-up duration. Stroke-related mortality remains high, ranging from 29.7% to 41.7%, with 10.3% to 47.4% of patients dying within three months of the event [218,219,220,221,222,223,224,225,226]. Predictive factors for stroke prior to transplantation include advanced age, diabetes mellitus, atrial fibrillation, and hypertension. Patient-specific factors, such as tobacco use, obesity, and prior cardiovascular events, further elevate risk. Post-transplantation, complications like graft failure or rejection are associated with a higher stroke incidence. These findings underscore the importance of rigorous cardiovascular risk management before and after transplantation to mitigate stroke-related morbidity and mortality.

Compared with waitlisted patients with ESRD, KT significantly reduces the incidence of stroke. Huang et al. [219] reported an incidence density of 5.34 per 1000 person-years in KTRs compared with 14.5 per 1000 person-years in ESRD patients, suggesting a substantial reduction. This benefit is likely due to the resolution of non-traditional and dialysis-related risk factors post-transplantation. However, potential selection bias exists, as KTRs are typically younger and have fewer comorbidities.

Lentine et al. [220] demonstrated that the cumulative three-year incidence of de novo cerebrovascular disease events post-transplantation was 6.8%, significantly lower than the adjusted three-year estimates of 11.8% in waitlisted patients and 11.2% in those with graft loss. Time-dependent regression analysis revealed that transplantation predicted a 34% reduction in cerebrovascular disease risk, while graft failure increased the risk by over 150%. Weng et al. [226] corroborated these findings, reporting a 60% reduction in overall stroke risk among the KTRs compared with the non-recipients, with ischemic and hemorrhagic stroke risks reduced by 48% and 74%, respectively. Regarding ischemic stroke mortality, Zhang et al. [224] found no significant difference in in-hospital mortality between the KTRs and a CKD-free population, whereas patients with CKD G5D faced markedly higher in-hospital mortality (adjusted odds ratio [aOR]: 2.04, 95% CI: 1.93–2.15). These results emphasize the protective effects of kidney transplantation on cerebrovascular outcomes and the detrimental impact of graft loss and dialysis dependence.

Stroke prevention strategies in KTRs are similar to those in CKD patients, focusing on managing traditional stroke risk factors, particularly atherosclerosis, using antiplatelet therapy and treating symptomatic severe carotid stenosis [215]. The NASCET study [227] demonstrated that CKD patients benefited more from carotid endarterectomy than non-CKD patients. However, the benefit of this procedure for asymptomatic carotid stenosis remains unclear in both CKD patients and KTRs due to the challenges of assessing surgical risks in these populations [228]. The SPACE-2 randomized controlled trial [229], with a five-year follow-up, found no benefit from carotid revascularization via stenting or endarterectomy. Additionally, an observational study reported a fourfold increase in in-hospital mortality associated with both techniques in ESRD patients [230].

The initiation of dialysis represents a particularly high-risk period, with stroke incidence peaking during this time [229]. While the mechanisms underlying this increased risk are not fully understood, it underscores the potential value of preemptive kidney transplantation as a preventative measure. However, current data on preemptive transplantation’s impact on cerebrovascular outcomes are sparse, highlighting a critical need for further research.

### 4.3. Management of Atrial Fibrillation in Kidney Transplant Recipients

AF accounts for approximately 25% of cardioembolic strokes [225]. Watschinger et al. identified pretransplant AF as a significant predictor of stroke in KTRs [5]. In the general population, AF is associated with an elevated risk of stroke and adverse outcomes including increased mortality, cognitive decline, and left ventricular dysfunction [231]. Anticoagulation is recommended for high-risk individuals with a CHADS-VASc score >2 (or >1 for men), a category encompassing many CKD patients who are particularly vulnerable to AF due to mechanisms like inflammasome pathway activation [232,233,234].

However, anticoagulation in ESRD patients remains controversial, as the bleeding risk often outweighs its benefits. The RENAL-AF trial highlighted this concern, reporting that major bleeding events occurred ten times more frequently than strokes or systemic embolisms in ESRD patients [235]. While direct oral anticoagulants (DOACs) are associated with a lower bleeding risk compared with vitamin K antagonists in ESRD patients [236], their overall benefit in terms of reducing embolic risk and mortality in this population remains debated [237]. Moreover, DOAC use is limited in dialysis patients on the kidney transplant waiting list due to the complexities of managing perioperative bleeding risks during transplantation [238]. Emerging approaches like left atrial appendage closure may provide an alternative for this unique population [239]. A recent meta-analysis demonstrated that DOACs are probably a safer option than vitamin K antagonists after kidney transplantation, with a similar efficacy [240].

A meta-analysis of eight cohort studies involving 137,709 KTRs provided key insights into the epidemiology and outcomes of AF in this population. Pre-existing AF had a prevalence of 7.0% (95% CI: 5.6–8.8%), while the incidence of post-transplant AF was 4.9% (95% CI: 1.7–13.0%). Despite transplantation, the burden of AF remained substantial, with a 2.5-fold increased risk of stroke and a 1.9-fold increased risk of mortality. Additionally, AF was associated with adverse post-transplant outcomes including a 1.5-fold increase in death-censored allograft loss [241].

These findings highlight the critical need for further research to guide the effective management of AF in kidney transplant recipients. Developing tailored strategies to balance thromboembolic prevention with bleeding risk will be essential for optimizing outcomes in this high-risk population.

### 4.4. Management of Peripheral Artery Disease in Kidney Transplant Recipients

The epidemiology of peripheral arterial disease (PAD) remains challenging to accurately define due to the polymorphic clinical presentation of chronic limb ischemia and intermittent claudication, which increases the likelihood of underdiagnosis. However, in high-income countries, the prevalence of PAD is estimated at approximately 5% among individuals aged 40–44 years and around 12% among those aged 70–74 years [242]. The prognosis for critical limb ischemia (CLI) is particularly grim, with an all-cause mortality rate of 22% (95% confidence interval [CI]: 12–33%) and a major amputation rate of 22% (95% CI: 2–42%) [243].

PAD is significantly more prevalent among individuals with CKD, with estimates ranging from 12% to 38%, even in mild to moderate CKD stages [244]. CKD patients face a higher risk of severe PAD complications including amputation. Despite its severity, PAD in CKD is frequently underdiagnosed and undertreated, leading to poor outcomes such as low amputation-free survival rates (~80% at one year) and a high risk of amputation in dialysis patients [245]. Diagnostic tools like the ankle-brachial index (ABI) have limitations in CKD due to vascular calcification, which may cause false-normal or elevated results. Performing ABI dynamically during a one-minute treadmill walking test can improve its diagnostic accuracy [246].

The pathogenesis of PAD in CKD involves both traditional risk factors (e.g., diabetes, hypertension) and non-traditional mechanisms such as vascular calcification, inflammation, impaired angiogenesis, uremic toxins, and microvascular disease. These mechanisms accelerate vascular aging and contribute to both atherosclerosis and medial arterial calcification, complicating diagnosis and treatment [244,247,248].

In KTRs, the risk of PAD is lower compared with dialysis patients. However, PAD remains associated with poor outcomes, including reduced allograft survival and higher mortality, with a reported threefold increase in the relative risk of death in affected KTRs [249]. Data from the U.S. Renal Data System highlight that symptomatic PAD at waitlisting or post-transplant amputation correlates with diminished allograft and patient survival [250,251]. Furthermore, Patel et al. demonstrated that even asymptomatic PAD, identified by a low ABI, is linked to a threefold higher risk of graft failure [252].

Despite these findings, routine screening for PAD in asymptomatic patients awaiting kidney transplantation remains controversial. While ensuring adequate vascular access is essential for successful anastomosis during transplantation, performing aortic bypass surgery is associated with high perioperative and postoperative mortality and morbidity in ESRD patients [253]. As a result, this procedure should be reserved for treating aneurysmal disease or symptomatic PAD, as per the current guidelines [254]. Conversely, evidence suggests that transplanting a kidney to the iliac artery does not significantly worsen ipsilateral lower extremity ischemia in adults with PAD [255,256]. Additionally, in high-risk populations, such as individuals with type 1 diabetes, proposing a simultaneous kidney-pancreas transplantation may help mitigate the development and progression of PAD [257].

These observations highlight the critical need to develop tailored diagnostic and therapeutic strategies to improve outcomes in the CKD and kidney transplant populations. A better understanding of PAD pathophysiology and management in these groups will ultimately lead to improved patient and graft survival.

## 5. Limitations

This narrative review had some inherent limitations. We used a general set of criteria to select studies, which was based on their overall quality and pertinence rather than a predefined protocol. Therefore, this may have introduced a selection bias despite our efforts to minimize it. Additionally, we aimed to describe the incidence and impact of post-transplant cardiovascular diseases, but most of the available data were derived from North American and European epidemiological studies. Our study may not adequately reflect the situation in other countries, such as Asian countries, or even peculiar European countries, due to the variations across populations. Although we aimed to provide a comprehensive overview, certain subjects on the topic may not have been comprehensively addressed due to the variation in the study designs and potential residual confounding bias in the observational studies such as treatment adherence or socioeconomic status. Obviously, this narrative review was subject to a risk of publication bias, since studies with statistically significant findings are more likely to be published. Future systematic reviews and meta-analyses are therefore needed for accuracy and to validate our data.

## 6. Conclusions

In conclusion, CVD remains a significant challenge for kidney transplant recipients, influencing both graft and patient survival. The complex interplay between traditional and transplant-specific risk factors necessitates a tailored approach to risk assessment, prevention, and management. While advancements in diagnostic tools and therapeutic strategies have improved outcomes, there are still gaps in our understanding of the optimal management of CVD in this population. Integrating novel biomarkers, refining screening algorithms, and exploring emerging therapies such as SGLT2 inhibitors and GLP-1 receptor agonists offer promising avenues for reducing cardiovascular complications. Future research should prioritize robust, large-scale studies to establish evidence-based protocols that can guide clinical practice and improve the quality of life for kidney transplant recipients. By addressing these challenges, we can move closer to optimizing the long-term outcomes and reducing the burden of cardiovascular morbidity and mortality in this vulnerable population.

## Figures and Tables

**Figure 1 diagnostics-15-00802-f001:**
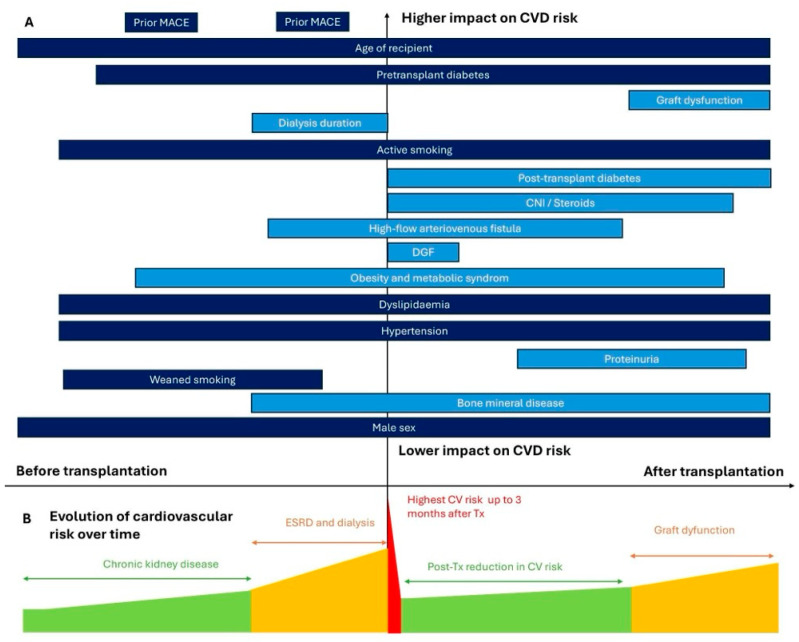
Evolution of classical and non-classical risk factors over time and their impact on CVD occurrence. (**A**) Distribution of cardiovascular risk factors based on their impact on cardiovascular disease (CVD) risk and their timing relative to transplantation. The vertical axis represents the magnitude of impact on cardiovascular risk, while the horizontal axis indicates the time before or after transplantation (zero = day of transplantation). Classical risk factors are shown in light blue, whereas non-classical risk factors are depicted in dark blue. (**B**) Evolution of cardiovascular risk over time. Periods of gradual increase in CVD risk are represented in green, periods of significant increase are shown in orange, and the phase with the highest impact on CVD risk is highlighted in red. Abbreviations: MACE, major adverse cardiovascular event; CVD, cardiovascular disease; CNI, calcineurin inhibitor; DGF, delayed graft function; ESRD, end-stage renal disease; CV risk, cardiovascular risk.

**Figure 2 diagnostics-15-00802-f002:**
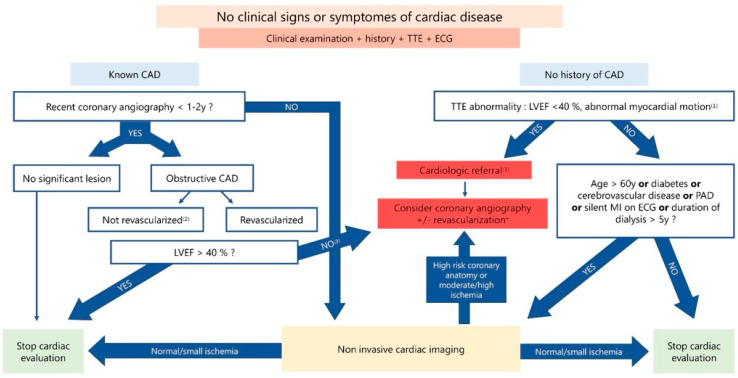
Simplified algorithm for cardiac screening in potential asymptomatic kidney recipients, adapted from Cheng et al. [90]. This algorithm does not apply to cases involving symptoms or clinical signs of cardiac involvement, which require specialized evaluation. Guideline-directed medical therapy (GDMT) remains indicated in all scenarios. (1) For non-ischemic abnormalities, pulmonary hypertension, or valvular disease: specific management tailored to the condition is necessary. (2) If GDMT proves inadequate, referral to a specialist is recommended, or reconsideration of transplant listing may be warranted. (3) If left ventricular ejection fraction (LVEF) remains < 40% despite optimal treatment, referral to a specialist or reconsideration of transplant listing is advised. Abbreviations: TTE, transthoracic echocardiography; ECG, electrocardiogram; CAD, coronary artery disease; LVEF, left ventricular ejection fraction; PAD, peripheral artery disease; MI, myocardial infarction; GDMT, guideline-directed medical therapy.

**Table 3 diagnostics-15-00802-t003:** Main cardiovascular and kidney adverse effects of maintenance therapies in kidney transplantation adapted from KDIGO 2009, Opalka et al. [168] and Elezaby et al. [168] Abbreviations: CTC, corticosteroid, CsA, cyclosporine A, TAC, tacrolimus, imTor, mTOR inhibitors, MMF, mycophenolate mofetil; Bela, belatacept, HTN, hypertension, GFR, glomerular filtration rate, ↓: protective, ↑: mild to moderate side effect, ↑↑: moderate to severe side effect.

Side Effects of Immunosuppressive Treatment	CTC	CsA	TAC	imTor	MMF	Belatacept
Post-transplantation diabetes melitus	↑↑	↑	↑↑	↑		↓
Dyslipidemia	↑	↑↑	↑	↑↑	↓	↓
Hypertension (HTN)	↑↑	↑↑	↑		↓	↓
Glomerular proteinuria		↓	↓	↑↑		
Kidney fibrosis		↑↑	↑↑	↑		↓
Cardiac hypertrophy, fibrosis	↑↑	↑↑	↑↑	↓	↓	↓
Vascular remodeling		↑	↑	↓	↓	↓

## Data Availability

Not applicable.

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
