# Peer review of "From Risk Assessment to Management: Cardiovascular Complications in Pre- and Post-Kidney Transplant Recipients: A Narrative Review"

_diagnostics, 2025, doi:10.3390/diagnostics15070802_

Round 1
Reviewer 1 Report (Previous Reviewer 1)
Comments and Suggestions for Authors
I appreciate your valuable work, but it appears that many articles may have been missed. However, I understand that this is not due to a systematic search strategy.
- My main concern is the omission of lifestyle recommendations and research results.
- If you are using figures from other articles, such as Figure 1, you should seek permission from the authors.
- Some subtitles, such as the one on line 233, should be bolded and separated for clarity.
- Additionally, the review should include its limitations, such as what aspects may be constrained due to the narrative nature of the writing.
Author Response
Reviewer 1:
I appreciate your valuable work, but it appears that many articles may have been missed. However, I understand that this is not due to a systematic search strategy.
We thank the reviewer for her/his valuable remarks. We acknowledge that we missed articles related to our research field. However, due to the broad scope of the review and its narrative style, we aimed to focus around the most pertinent guidelines and articles.
1. My main concern is the omission of lifestyle recommendations and research results.
A part concerning the role of lifestyle modifications was added (Line 643 to 690), where we discuss the general guidelines applicable to kidney transplant recipients, as well as significant research results about the impact of dietary changes and physical activity.
“2.3. The Role of Lifestyle Modifications
According to the KDIGO 2009 Guidelines, kidney transplant recipients should follow a “healthy lifestyle, with exercise, proper diet, and weight reduction as needed” [123]. These recommandations are obviously adapted from those concerning the general population: According to the 2019 ACC/AHA guidelines, lifestyle factors can indeed influence and reduce cardiovascular risk in the general population [134]. Concerning nutrition and diet, patients should privilege vegetables such as fruits, legumes, nuts and whole grain, as well as fish intake. Saturated fat should be replaced by mono- or polyinsaturated fats. The amount of cholesterol and salt consumption should be reduced. A high potassium intake can also be encouraged in the general population but should be carefully evaluated in kidney transplant recipients. Finally, as part of the healthy diet, several intakes should be minimized, such as trans fats, processed meats, refined carbohydrates ans sweetened beverages. Patients also should be encouraged, if able, to routinely perform at least 150 minutes of moderate intensity or 75 minutes high intensity physical activity per week. Lifestyle changes also imply weight loss, tobacco weaning and alcohol avoidance. They participate in a global reduction of cardiovascular disease in the general population, mainly through the reduction of blood pressure and LDL-Cholesterol levels, and through the improvement of glycemic control. They also participate in reducing BMI and are the baseline treatment for obesity [134]. The 2019 KDIGO guidelines also precise that, up to date, there is no reason to believe that a healthy liestyle and a proper diet do not prevent cardiovascular complications as in the general population [123].
Several randomized control trials analysed the impact of physical acticity in the field of kidney transplantation, but are globally underpowered to show a real difference in cardiovascular risk factors improvement. Most of them are centered on the improvement of physical capacity and kidney function. A meta-analysis by Zhang et al concerning exercice intervention in kidney transplant recipients and including 16 randomized control trials revealed a positive impact of exercice on renal function and physical performance, with a lower creatinine and an improved VO2 peak and 6-minutes walk test [135]. However, a randomized control trial of 96 patients did not show significant improvement in cardiovascular risk, evaluated according to Framingham equations, in the exercice arm in the first year after transplantation [136].
Concerning the impact of dietary changes, they were associated in several clinical trials to a better kidney function and overall survival. In a randomized clinical trial including 632 kidney transplant recipients, mediterranean diet was associated with better graft function [137]. Another trial including 632 recipients determined that the DASH diet was associated with better graft function and lower all-cause mortality [138]. However, a recent systematic review about dietary interventions for body weight management did not demonstrate any effect due to a lack of significant studies in chronic kidney disease [139] and kidney transplant recipients [140]. Unfortunately, impact of dietary changes on specific cardiovascular outcomes are rarely depicted in the litterature. In summary, if some specific diet styles are associated with better outcomes, the impact of dietary interventions remains to be proven.
The proper impact of physical activity and dietary changes remains unclear on specific cardiovascular risk factors and outcomes in kidney transplant recipients, mostly due to unadapted designs in clinical studies. However, due to their large beneficial impact in the general population, and based on the available litterature, we can conclude that lifestyle changes are formally indicated in all kidney transplant recipients, regardless of their cardiovascular risk.”
2. If you are using figures from other articles, such as Figure 1, you should seek permission from the authors.
Figure 1 is an original figure designed by the authors.
3. Some subtitles, such as the one on line 233, should be bolded and separated for clarity.
We improved visibility and harmonized the subtitles. For example, on line 233, the subtitle was italicized.
4. Additionally, the review should include its limitations, such as what aspects may be constrained due to the narrative nature of the writing.
We added the following paragraph to address the limitations of the narrative review, from line 1169 to line 1183:
“5. Limits.
This narrative review has inherent limitations. We did use a general set of criteria to select studies, based on their overall quality and pertinence rather than a predefined protocol. Therefore, this may have introduced a selection bias despite our efforts to minimize it. Additionally, we aimed to describe the incidence and impact of post-transplant cardiovascular diseases but most of the available data are derived from North-American and European epidemiological studies. Our study may not reflect adequately the situation in other countries, such as Asian countries, or even peculiar European countries, due to the variations across populations. Although we aimed to provide a comprehensive overview, certain subjects about the topic may not have been comprehensively addressed due to the variation in study designs, and to potential residual confounding bias in observational studies, such as treatment adherence or socioeconomic status. Obviously, this narrative review is subject to a risk of publication bias, since studies with statistically significant findings are more likely to be published. Future systematic reviews and meta-analyses are therefore needed to precise and validate our data.”
Reviewer 2 Report (New Reviewer)
Comments and Suggestions for Authors
I considered the manuscript entitled “From Risk Assessment to Management: Cardiovascular Complications in Pre- and Post-Kidney Transplant Recipients: a narrative review “ byThomas Beaudrey, et al, that is intended to be published in Diagnostics Journal.
The narrative is comprehensive, robust and well-focused. The approach to “We prioritized studies including observational data or clinical trials related to cardiovascular risk factors and diseases in the peculiar population of kidney transplant recipients” give special rationality for clinicians.
In the first paragraph of introduction, I would add nephroangioesclerosis appearing in the graft as part of cardiovascular vascular disease, apart from coronary, periphery and stoke. I should be searched additional literature to help understand the fate of post-transplant allograft in terms of CVD, later in the narrative
“its incidence has significantly decreased [11,12], likely due to advancements in primary and secondary prevention strategies as well as improvements in treatment”. I agree with you but some comments should be added mentioning that this is despite the age of acceptance of receptors has progressively increase.
Figure 1 is highly illustrative and didactic
Chapter 2. Management of Post-Transplant Cardiovascular Risk 2.1. Treatment of Traditional Risk Factors in Kidney Transplant Recipients, and chapter 2.5.2. Off-Targets Adverse Effects of Maintenance Therapy: A Personalized Strategy are somewhat repetitive and redundant. It should be rearranged.
An impressive number of literature citations, as corresponds with a comprehensive review
Author Response
Reviewer 2:
I considered the manuscript entitled “From Risk Assessment to Management: Cardiovascular Complications in Pre- and Post-Kidney Transplant Recipients: a narrative review” by Thomas Beaudrey, et al, that is intended to be published in Diagnostics Journal.
The narrative is comprehensive, robust and well-focused. The approach to “We prioritized studies including observational data or clinical trials related to cardiovascular risk factors and diseases in the peculiar population of kidney transplant recipients” give special rationality for clinicians.
We thank the reviewer for her/his valuable remarks.
In the first paragraph of introduction, I would add nephroangioesclerosis appearing in the graft as part of cardiovascular vascular disease, apart from coronary, periphery and stoke. I should be searched additional literature to help understand the fate of post-transplant allograft in terms of CVD, later in the narrative
We added a mention about post-transplant nephroangiosclerosis in the introduction, from line 38 to line 42:
“The spectrum of post-transplant CVD covers coronary artery disease, stroke and peripheral artery disease, with some peculiar complications, such as early post-transplant coronary artery disease, undetected despite pretransplant screening, and accelerated arteriosclerosis occurring in the transplant.”
The following paragraph was added from line 124 to line 135.
“Accelerated arteriosclerosis should also be considered as a part of post-transplant cardiovascular disease. Arteriosclerosis in kidney transplants is defined as a vascular fibrous intimal thickening, according to the Banff classification [36], progressing after kidney transplantation and distinct from baseline donor lesions [37]. The pathophysiology of transplant arteriosclerosis remains unclear but probably results from two main mechanisms: hypertension and other cardiovascular risk factors can indeed induce this type of lesions [38], but the most recent hypothesis incriminates alloimmune response related to donor-specific antibodies. In this study by Loupy et al [39], risk factors for arteriosclerosis were donor parameters, inducing baseline lesions, cold ischemia time, recipient hypertension requiring medical therapy and circulating donor-specific antibodies. In the same study, transplant arteriosclerosis was associated with graft failure and cardiovascular events, especially in the presence of donor-specific antibodies.”
“its incidence has significantly decreased [11,12], likely due to advancements in primary and secondary prevention strategies as well as improvements in treatment”. I agree with you but some comments should be added mentioning that this is despite the age of acceptance of receptors has progressively increase.
We agree with the reviewer on this point. We modified this sentence in the following way, line 72 to 75:
“Even today, CVD remains the leading cause of death in this population [9,10]; however, its incidence has significantly decreased [11,12], likely due to advancements in primary and secondary prevention strategies as well as improvements in treatment, and despite the older median age at the time of waitlisting and transplantation”
Figure 1 is highly illustrative and didactic
We thank the reviewer for her/his remark.
Chapter 2. Management of Post-Transplant Cardiovascular Risk 2.1. Treatment of Traditional Risk Factors in Kidney Transplant Recipients, and chapter 2.5.2. Off-Targets Adverse Effects of Maintenance Therapy: A Personalized Strategy are somewhat repetitive and redundant. It should be rearranged.
In Part 3, we discuss the management of post-transplant cardiovascular risk, addressing several specific parts:
3.1, 3.2 and 3.3 are based on general guidelines about risk factors and lifestyle changes.
3.4 and 3.5 are related on specific nephroprotective treatments (RAS blockers, SGLT2 inhibitors…)
3.6 and 3.7 are related to two specific factors, i.e. immunosuppressive treatment and vascular access management.
The reviewer points out that “3.1. Treatment of Traditional Risk Factors in Kidney Transplant Recipients” and and “3.6.2. Off-Targets Adverse Effects of Maintenance Therapy: A Personalized Strategy” are redundant. However, immunosuppressive treatment is not addressed in 3.1 nor in 3.2 part and we believe that it deserves its own part, given its inherent complexity and broad interest.
An impressive number of literature citations, as corresponds with a comprehensive review
We thank the reviewer for her/his remark.
Reviewer 3 Report (New Reviewer)
Comments and Suggestions for Authors
The Authors aimed to outline post-transplant cardiovascular risk and disease, describe the epidemiology of cardiovascular events, focus on traditional and specific cardiovascular risk factors, and screen for coronary artery disease before transplantation. They then described their management with cardiorenal medical therapies, immunosuppression, and vascular access management and concluded with the management of the main cardiovascular diseases after transplantation.
The literature search was not systematic; studies were selected based on subjective evaluations of relevance, methodology, and sample size.
In the epidemiological section, the Authors highlight the national/continental differences in the occurrence of CVD both before and in the early post-transplant period. I suggest extending the analysis to the general population since an increased prevalence of CVD in the population necessarily determines an increased prevalence in waiting list patients.
The description of typical and atypical risk factors is synthetic but exhaustive.
The description of pre-transplant CVD assessment is comprehensive and well-structured.
I suggest a more critical approach in the paragraph describing the usefulness of SGLT2i in kidney transplant patients, despite the causal approach of the cited study. I underline that in CKD not-transplanted patients, all the RCTs are based on the association ACE-i/ARBs + SGLT2i vs ACE-1/ARBs alone.
The other paragraph not cited in the review are thorough.
Author Response
Reviewer 3:
The Authors aimed to outline post-transplant cardiovascular risk and disease, describe the epidemiology of cardiovascular events, focus on traditional and specific cardiovascular risk factors, and screen for coronary artery disease before transplantation. They then described their management with cardiorenal medical therapies, immunosuppression, and vascular access management and concluded with the management of the main cardiovascular diseases after transplantation.
The literature search was not systematic; studies were selected based on subjective evaluations of relevance, methodology, and sample size.
We thank the reviewer for her/his general assessment of the manuscript.
In the epidemiological section, the Authors highlight the national/continental differences in the occurrence of CVD both before and in the early post-transplant period. I suggest extending the analysis to the general population since an increased prevalence of CVD in the population necessarily determines an increased prevalence in waiting list patients.
We agree and added the following sentence, from line 78 to line 80:
“These discrepancies are mainly attributable to inherent differential cardiovascular risk between high-income countries, with a CVD-related mortality consistently higher in the United States”
Reference: Acosta E, Mehta N, Myrskylä M, Ebeling M. Cardiovascular Mortality Gap Between the United States and Other High Life Expectancy Countries in 2000–2016. The Journals of Gerontology: Series B [Internet]. 2022 May 1 [cited 2025 Mar 17];77(Supplement_2):S148–57. Available from: https://doi.org/10.1093/geronb/gbac032
The description of typical and atypical risk factors is synthetic but exhaustive.
The description of pre-transplant CVD assessment is comprehensive and well-structured.
We thank the reviewer for her/his valuable remark.
I suggest a more critical approach in the paragraph describing the usefulness of SGLT2i in kidney transplant patients, despite the causal approach of the cited study. I underline that in CKD not-transplanted patients, all the RCTs are based on the association ACE-i/ARBs + SGLT2i vs ACE-1/ARBs alone.
We agree and added the following sentences from line 771 to line 777.
“However, these results should be interpreted with caution, since only 44% of the par-ticipants in this study were treated using RAAS inhibitors. All randomized controlled trials performed in non-transplanted CKD patients [121,122] compared the RAAS blockers-SGLT2 inhibitors association to RAAS blockers alone and a similar design should be applied to the transplant population. It is rather reassuring that no significant difference in the effect size of SGLT2 inhibitors was described in this study, between subgroups with or without RAAS inhibitors treatment.”
The other paragraph not cited in the review are thorough.
We thank again the reviewer.
Round 2
Reviewer 2 Report (New Reviewer)
Comments and Suggestions for Authors
none
This manuscript is a resubmission of an earlier submission. The following is a list of the peer review reports and author responses from that submission.
Round 1
Reviewer 1 Report
Comments and Suggestions for Authors
1. While the background and aim of the narrative review are introduced in the abstract, I recommend adding an Introduction and/or Methods section. This section should describe how articles were selected, even if the selection process was not systematic. It would also be beneficial to explain the rationale behind the thematic organization of the review.
2. Regarding the thematic organization, specifically themes 3 and 4, I found the transition between risk assessment and management sections confusing. It was difficult to discern when the discussion shifted from risk assessment to management, especially as the article title suggests both are covered. Clarifying and possibly integrating these sections could improve the flow and readability of the review.
3. Lastly, the inclusion of lifestyle interventions such as physical activity seems underrepresented as a management strategy. While lifestyle management is mentioned, it appears only marginally related to other management strategies like cholesterol management. I suggest giving more emphasis to lifestyle interventions, detailing how they fit into the overall management approach to provide a comprehensive view of the treatment landscape.
Reviewer 2 Report
Comments and Suggestions for Authors
THough the authors analize important topic, but I absolutely missed the methodology, research question. It's like a paragraph in the book but not the scientific article.. Rewriting the known data.